# Bee Venom: From Venom to Drug

**DOI:** 10.3390/molecules26164941

**Published:** 2021-08-15

**Authors:** Abdelwahab Khalil, Basem H. Elesawy, Tarek M. Ali, Osama M. Ahmed

**Affiliations:** 1Entomology Division, Zoology Department, Faculty of Science, Beni-Suef University, Beni-Suef 62521, Egypt; akhalil1980@gmail.com; 2Department of Clinical Laboratory Sciences, College of Applied Medical Sciences, Taif University, P.O. Box 11099, Taif 21944, Saudi Arabia; basemelesawy2@gmail.com; 3Department of Pathology, Faculty of Medicine, Mansoura University, Mansoura 35516, Egypt; 4Department of Physiology, College of Medicine, Taif University, P.O. Box 11099, Taif 21944, Saudi Arabia; tarek70ali@gmail.com; 5Department of Physiology, Faculty of Medicine, Beni-Suef University, Beni-Suef 62521, Egypt; 6Physiology Division, Zoology Department, Faculty of Science, Beni-Suef University, Beni-Suef 62521, Egypt

**Keywords:** bee venom, inflammatory and chronic diseases, nanoparticles, bee venom products

## Abstract

Insects of the order Hymenoptera have a defensive substance that contains many biologically active compounds. Specifically, venom from honeybees (*Apis mellifera*) contains many enzymes and peptides that are effective against various diseases. Different research papers stated the possibility of using bee venom (a direct bee sting or in an injectable form) in treating several complications; either in vivo or in vitro. Other reports used the active fractions of bee venom clinically or at labratory scale. Many reports and publications have stated that bee venom and its constituents have multiple biological activities including anti-microbial, anti-protozoan, anti-cancer, anti-inflammatory, and anti-arthritic properties. The present review aims to refer to the use of bee venom itself or its fractions in treating several diseases and counteracting drug toxicities as an alternative protocol of therapy. The updated molecular mechanisms of actions of bee venom and its components are discussed in light of the previous updated publications. The review also summarizes the potential of venom loaded on nanoparticles as a drug delivery vehicle and its molecular mechanisms. Finally, the products of bee venom available in markets are also demonstrated.

## 1. Introduction

Apitherapy is an alternate therapy that relies on the usage of honeybee products, most importantly bee venom, for the treatment of many human diseases [1,2,3,4]. The curative effects of honey and other bee products are stated in different religious books as Veda, the Bible, and the holy Quran [5,6]. God almighty has confirmed the truth when he told in the holy Qur’an the following:

“And your Lord inspired to the bee,” take for yourself among the mountains, houses and among the trees and [in] that which they construct (68). Then eat from all the fruits and follow the ways of your Lord laid down [for you]. their emerges from their bellies a drink, varying in colors, in which there is healing for a people. Indeed, in that is a sign for a people who give thought (69). Surat Al-Nahl.

For thousands of years, humans have used honeybee products including honey, propolis, and venom from the European honeybee *Apis mellifera* as medicines; the use of these honeybee’s products for therapy of diseases is known as apitherapy. The medicinal use of bee venom dates back to ancient Egypt and Greece, and has been practiced in China for 3000–5000 years [7]. The ancient Greek doctor Hippocrates used bee venom for therapeutic purposes [8]. In more modern times, interest in the effects of bee venom was renewed in 1868 by the Russians Lokumski and Lubarski, who published a work named “Bee venom, a remedy” [9]. Recently, physicians and licensed apitherapists have been using honeybee venom for treating patients that suffer from chronic or autoimmune diseases. Both clinical trials and lab testing confirmed that honeybee venom is an excellent form of biotherapy. Honeybee venom either fights off inflammation and destruction of connective tissues (as in the case of rheumatism and arthritis) or it returns activity and mobility by supporting the natural body defense (as in the case of multiple sclerosis and lupus) [1,10,11]. Lately, the venom has been also evaluated for treating different cancer types [12,13,14]. In addition to its therapeutic effects, the literature stated that the bee venom might decrease the adverse effects of other types of medications and conventional drugs [15,16,17].

This review aims to give a comprehensive updated account of bee venom compositions, benefits, and applications in experimental and clinical medicines. The review will also explain how this venom can aid in treating many human diseases and counteracting the adverse side effects of various drugs. Furthermore, the review also discusses the importance of the use of polymers and their nano-forms as carriers for the bee venom.

## 2. Hymenopteran Insects

Bees, ants, and wasps belonging to the entomological order: Hymenoptera can cause a toxicity to humans through envenomation. Some hymenopterans are solitary, but others are beneficial such as pollinating bees and parasitoids. Hymenopteran species that can affect humans are stinging insects, which can use their stinging tool as a vindicatory or aggressive arm. Those that coexist in colonization cause dangerous stings. Their toxins often induce pain against the predators. One of the interesting patterns of hymenopterans is stinging. Different factors induce stinging response such as visual signals and vibrations. Moreover, vibrations of the nests or annoyances of workers are factors that induce stinging or envenoming [18].

Hymenopteran venoms are highly complex combinations of salts, organic elements (i.e., amino acids, alkaloids), and neurotransmitters [19,20]. The venoms’ compositions vary greatly between hymenopteran species and the concentration of some constituents, and vary in species of the same genus as well [21,22]. Bee venoms contain some of the same compounds as wasp venoms, such as adrenaline, noradrenaline, dopamine, serotonin, histamine, hyaluronidase, phospholipases B (PLBs), and phospholipases A_2_ (PLA_2_s), while only bee venoms contain apamin, melittin, and mast cell-degranulating peptide (MCD) [23].

## 3. Bee Venom

Bee venom (api-toxin) is secreted by a gland located in the abdominal cavity of the bees (*Apis mellifera* L.). It is an odorless and transparent acidic liquid that bees often use as a defense tool against predators. Honeybee venom (commercially known as Apitox or Apitoxin) is a combination of different compounds. As reported by many publications, bee venom contains several active molecules such as peptides and enzymes including melittin (a major component of bee venom), apamin, adolapin, mast cell degranulating peptide, and enzymes (phospholipase A2 and hyaluronidase), as well as non-peptide components, such as histamine, dopamine, and norepinephrine [3,24,25]. The main constituents are melittin, which accounts for about 50 percent of dry venom, and phospholipase A_2_ (PLA2), which accounts for approximately 12 percent [26].

In classical medicine, bee venom and bee-derived toxins were used for treating chronic inflammatory disorders, as they have different effects such as anti-arthritic, anti-cancer, and pain killer potencies [24,26,27]. In bee sting therapy, the honeybees go directly to the target point via stinger, while in bee venom therapy, the lyophilized venom (collected from bees then lyophilized) is injected directly by different doses in situ [28,29]. Injecting bee venom has the ability to treat different illnesses such as autoimmune disorders (rheumatoid arthritis, psoriasis, and so on), neurological disorders, chronic inflammations, pain, skin diseases, and microbial infections [2,8,29].

### 3.1. Physical Properties, Chemistry, and Pharmacology of Honeybee Venom

Honeybee venom is a clear liquid with bitter taste, fragrant sharp, 1.13 specific gravity, and 4.5–5.5 pH [30,31]. When the honeybee venom comes into contact with the air, it quickly dries and crystallizes [32]. Dried venom becomes a light yellow colour and some commercial preparations are brown, thought to be because of oxidation of some of the venom proteins. It is soluble in water and insoluble in alcohol and ammonium sulfate. Bee venom contains a number of very volatile compounds, which are easily lost during collection [28].

Honeybee venom has been reported to contain a mixture of many components. The category of such components includes proteins acting as enzymes such as phospholipase A2, phospholipase B, acid phosphomonoesterase, hyaluronidase, phosphatase, and lysophospholipase, as well as smaller proteins and peptides such as melittin, apamin, adolapin, tertiapin, and secapin. Other components include phospholipids and physiologically active amines such as histamine, dopamine, and noradrenaline. Further components are amino acids, sugars such as glucose and fructose, pheromones, and minerals such as calcium and magnesium. The major constituent of honeybee venom is melittin, which is composed of 26 amino acids and represents 40–50% of the dry venom [8,12,33].

The pharmacology of bee venom has been studied through in vivo and in vitro studies [1,4,12,17,34,35,36,37]. The bee venom has multiple diverse pharmacological effects such as anti-mutagenic [37], anti-nociceptive [37], radioprotective [37], anti-hepatotoxic [17], cytoprotective [17], anti-oxidant [4,17], anti-microbial [4], anti-viral [4], anti-inflammatory [4,17], neuroprotective [4], anti-arthritic [1], anti-metastatic [12], and anti-tumor [12] effects (Figure 1).

### 3.2. In Vitro and Pre-Clinical Studies

Bee venom has been investigated by several in vitro and pre-clinical studies that have evidenced an array of its multiple biological activities. 

Bee venom has been revealed to have anti-oxidant efficacies both in vitro and in vivo. Concerning in vitro studies, the anti-oxidant activity of bee venom was evaluated by various methods including 2,2-diphenyl-1-picrylhydrazyl (DPPH) scavenging capacity, ferric reducing/antioxidant power (FRAP), 2,20-azinobis 3-ethylbenzothiazoline-6-sulfonic acid (ABTS) free radical scavenging power, thiobarbituric acid reactive substances (TBARS) inhibition, and β-carotene bleaching inhibition assays [4,38,39]. The results of these studies revealed the anti-oxidant potentials of bee venom, although it was not linked to a certain individual constituting compound. Somwongin et al. [39] compared the anti-oxidant effects of different species of *Apis* and found that all the venom extracts showed inhibition of DPPH, whereas the highest activity was displayed by *Apis dorsata* followed by *Apis mellifera*.

Regarding in vivo studies, El-Hanoun et al. [40] injected into rabbits 0.1, 0.2, and 0.3 mg per rabbit by subcutaneous administration twice/week for 20 weeks. The results showed an increase in glutathione-S-transferase (GST) activity and glutathione (GSH) content and a decrease in malondialdehyde (MDA) and TBARS levels in the treated rabbits, confirming the anti-oxidant effect of honeybee venom. Furthermore, Mohamed et al. [41] found that rats with induced gastric ulcer treated with bee venom and acetylsalicylic acid attenuated lipid peroxidation. Ahmed et al. [1] reported that the deterioration in the antioxidant defense system in complete Freund’s adjuvant (CFA)-induced arthritic rats was significantly improved as a result of bee venom administration. The improvement in the antioxidant defense system was manifested by the decrease in liver lipid peroxidation and increase in glutathione content and activities of antioxidant enzymes including glutathione peroxidase and GST.

In addition to its anti-oxidant activity, bee venom has anti-inflammatory properties. An anti-inflammatory effect of bee venom was reported in CFA-induced arthritis in rats [1] and in phthalic anhydride-induced atopic dermatitis animal model [42]. The anti-inflammatory effects were manifested by a decrease in the elevated serum interleukin-2 (IL-2) and interleukin-12 (IL-12) levels and an increase in interleukin-10 (IL-10) level in CFA-induced arthritic rats as a result of treatment with bee venom [1]. The anti-inflammatory activity bee venom may be attributed to its main component, melittin. This compound was reported to have anti-inflammatory effects against acne vulgaris, neuro-inflammation, amyotrophic lateral sclerosis, arthritis, and liver inflammation [43]. The treatment with melittin modulates Toll-like receptors (TLRs) pathways’ activation and inhibits the expression of inflammatory cytokines. Melittin was found to suppress the activation of nuclear factor kappa B (NF-κB) p65 and inhibit the protein 38 mitogen-activated protein kinase (p38 MAPK) signal in vitro [44]. Therefore, the anti-inflammatory activity may be mediated by NF-κB and p38 signaling pathways. In vivo, melittin also showed anti-inflammatory properties by the modulation of NF-κB and activator protein 1 (AP-1) transcription factors (Figure 2) [44]. It was reported that melittin suppresses signaling pathways of Toll-like receptor type 2 (TLR2), Toll-like receptor type 4 (TLR4), cluster differentiation 14 (CD14), NF-κB essential modulator (NEMO), and platelet-derived growth factor receptor beta (PDGFRβ). By affecting these signaling pathways, melittin decreases activation of NF-κB protein 38 (p38), extracellular signal-regulated kinase 1/**2** (ERK1/2), protein kinase B (PKB), Akt, or phospholipase Cγ1 (PLCγ1), as well as translocation of NF-κB into the nucleus (Figure 2). This deactivation results in decreased inflammation in skin, arota, joint, liver, and neuronal tissue (Figure 2) [45]. In another signaling pathway, bee venom (or melittin) inhibits the release of inhibitor of NF-κB (IκB) through the inhibition of IκB kinases (IKKs). Such inhibition could be owing to an interaction between the SH group of IκB kinase α (IKKα) and IκB kinase β (IKKβ) with bee venom or melittin molecule, which results in NF-κB deactivation, and thus decreases the generation of inflammatory mediators. Bee venom as well as melittin may also interact directly with p50 of NF-κB and, thereby, suppress the translocation of p50 into the nucleus (Figure 2) [46]. A variety of extracellular signals including TNF-α (canonical pathway) can activate the enzyme IκB kinase (IKK), which results in activation of NF-κB and the release of p65/p50 heterodimers. The translocation of NF-κB p65/p50 heterodimers into the nucleus allows them to bind to their specific sequences of DNA and promotes NF-κB target genes involved in inflammatory responses [47]. The bee venom and melittin may produce their anti-inflammatory effects by affecting the canonical pathway of NF-κB through the inhibition of TNF-α receptors 1 and 2 (TNFR1 and TNFR2) (Figure 2) [48].

Bee venom and its main component, melittin, have inhibitory effects on cancer cell growth in prostate cancer. The down-regulation of anti-apoptotic gene products such as B-cell lymphoma 2 (Bcl-2) (Figure 3), X-linked inhibitor of apoptosis (XIAP), inducible nitric oxide synthase (iNOS), and cyclooxygenase-2 (COX-2) may be implicated to mediate the anti-cancer effects [49,50]. Remarkably, the crude bee venom and melittin have revealed anti-tumor activities against various cancer cell lines including breast, liver, leukemia, lung, melanoma, and prostate cancer cell lines [51,52,53,54]. The anti-cancer effects of bee venom and its major component melittin may be explained in several ways. Firstly, the anti-tumor effect may be mediated via induction of extrinsic and intrinsic pathways of apoptosis (Figure 3). It has been reported that bee venom and melittin increased the expression and levels of various proapoptotic and apoptotic mediators such as cytochrome C (Cyt C), protein 53 (p53), Bcl-2-associated X protein (Bax), Bcl-2 homologues antagonist/killer (Bak), caspase-3, caspase-9, and different types of death receptors, and they reduced anti-apoptotic mediator Bcl-2 [12,50,55]. The anti-tumor effects of bee venom and melittin may also be mediated via inhibition of NF-κB and calcium–calmodulin complex (Ca^2+^/CaM) signaling pathways. This inhibition, in turn, results in a decrease in tumor growth and survival through suppression of cell proliferation, angiogenesis, anti-apoptosis, invasion, and metastasis (Figure 3) [12]. Bee venom was reported to increase the production of ROS from mitochondria, which induce intrinsic apoptosis. In addition, bee venom and melittin increased the production of apoptosis-induced factor (AIF) and endonuclease G (EndoG), which are mitochondrial proteins that activate caspase-independent apoptosis [56,57].

Bee venom has been reported to have anti-microbial activity, which was attributed to two of its major components, melittin and PLA2. Such compounds may exert their anti-microbial actions against bacteria by producing pores through their membranes, leading to their damage and then lysis [3,4,58]. According to Pucca et al. [59], these bee venom toxins may derive a synergistically enhanced toxicity via formation of toxin complexes by hetero-oligomerization. The enhanced bacterial lysis may be mediated via formation of melittin-PLA_2_ hetero-oligomers on phospholipid bilayer membranes. When melittin combines with PLA_2_s, melittin is more likely to easily bind to the neutral outer membrane on the lipid binding surface of the molecule, resulting in synergistically enhanced lysis. This binding is possibly mediated via electrostatic attraction between basic amino acid residues of melittin and the phosphate group of phosphatidylcholine (a major constituent of cell membranes). The melittin–phospholipid binding enables melittin oligomerization, formation of membrane pores, and lysis [23,60]. The anti-viral effect of bee venom has been reported with interesting findings both in vivo and in vitro. In the in vitro investigations, bee venom and melittin have significant anti-viral effects against non-enveloped viruses (coxsackie virus and enterovirus-71) and several enveloped viruses (vesicular stomatitis virus, influenza A virus, herpes simplex virus, and so on) [61]. In vivo, melittin protected mice that were exposed to lethal doses of influenza A H1N1 virus. Although the precise mechanism of action by which bee venom and melittin act as antiviral agents remains unclear, it has been confirmed that bee venom interacts directly with the viral surface. Moreover, bee venom and its components can stimulate type I interferon (IFN) and, therefore, suppress viral replication in the host cell [62]. 

In addition to the previous literature, Sig et al. [63], Bogdanov [8], Zhang et al. [2], and Oršolić [12] stated that bee venom has potent in vivo effects against arthritis, Parkinson’s and Alzheimer’s diseases, and cancer.

### 3.3. Clinical Applications

By considering the wide range of biological properties of bee venom, it would not be surprising that its use could reach therapeutic purposes for human diseases. Bee venom therapy, like many other complementary and alternative medicine approaches, has been used for treating many diseases [4,28].

Before the application of injections by syringes, honeybee venom was always supplied directly from honeybees via the honeybees’ stingers [64,65]. Nowadays, in some conditions, honeybee venom is still supplemented by such traditional method. The live honeybee is smoothly held with tweezers or some other tools by the person administering the honeybee venom, who then puts the honeybee on the part of the patient’s body to be treated, at which point the honeybee reflexively stings. It worth mentioning here that, when the honeybees sting other insects, they do not die. Nevertheless, the honeybee dies most of the time when it stings the human body. After stinging the human body, the honeybee cannot pull the stinger back out. It leaves behind not only the stinger, but also part of its digestive tract, plus muscles and nerves. This massive abdominal rupture is what kills the bee [66].

Depending on the condition, the treatment schedule can vary. In this regard, bee venom can be administered by different therapy methods, which include direct sting of the bee, bee venom injection, or bee venom acupuncture (also called apitherapy). Although bee venom administered via a syringe is recommended rather than directly from the honeybee, most studies and practices use bee venom acupuncture owing to the bioactivity coupled to mechanical stimulation of acupuncture [2,4,65].

In clinical settings, the bee venom injection at acupoints has been reported to be effective for the treatment of clinical disorders such as Parkinson’s disease, neuropathic pain, Alzheimer’s disease, intervertebral disc disease, spinal cord injury, musculoskeletal pain, arthritis, multiple sclerosis, skin disease, and cancer [67]. The effectiveness of bee venom injection at acupoints possibly results from its anti-inflammatory, anti-nociceptive, and anti-apoptosis effects.

Regarding the therapeutic potential of bees, a study performed in humans demonstrated that bee venom acupuncture showed efficacy as adjuvants in Parkinson’s disease treatment when adults were stimulated on 10 acupuncture points, twice a week for 8 weeks [68].

According to a clinical trial of Hauser et al. [10], it was found that therapeutic bee venom injections on patients with multiple sclerosis are effective in decreasing a patient’s functional debilitation caused by the disease. The patients showed significant ameliorations in balance, coordination, bladder and bowel control, upper- and lower-extremity strength, fatigue, endurance, spasticity, and numbness over the 12-month trial using bee venom treatment. Statistically, significant improvements were seen in walking, stair climbing, car transfers, bed transfers, toilet transfers, bathtub transfers, and bed positioning. In another clinical trial, Wesselius et al. [69] revealed that treatment with bee venom in patients with relapsing multiple sclerosis did not reduce disease activity, disability, or fatigue, and did not improve quality of life.

Liu et al. [70] reported that combined bee venom therapy application with other medications is more efficient than simple use of medication in treating rheumatoid arthritis. They also stated that, when bee sting therapy was applied, the doses of Western medicines may be decreased, and the relapse rate becomes lower.

Figure 4 [71,72,73,74,75,76,77,78,79,80,81,82,83,84] summarizes the therapeutic effects and mode of actions of bee venom in various diseases and conditions. The effects and actions in the figures are supported by references. 

Although there are many convincing results regarding the potential use of bee venom, more specifically melittin, against various types of cancer, its applicability to humans remains very challenging owing to its non-specific cytotoxicity [85].

In spite of wide-range therapeutic potentials of bee venom, there is a parallel reality of potential side effects or allergic reactions linked to bee venom administration [86].

#### 3.3.1. Allergy Testing, Directions for Patients, and Contraindications 

Systemic-allergic bee sting reactions have been demonstrated in up to 3.4% of children and up to 7.5% of adults. These allergic reactions can be categorized to be mild and restricted to the skin or moderate to severe with a risk of life-threatening anaphylaxis. Thus, before applying bee venom therapy, an allergy test should be conducted. The only treatment to eliminate further systemic sting reactions is venom immunotherapy, which is indicated to prevent further moderate-to-severe systemic sting reactions in venom-allergic children and adults [87].

For allergy testing, about 0.05 mL (1.0 mg of pure lyophilized venom dissolved in 1.0 mL of physiological saline) is injected in the flexor surface of the forearm of the patient. It is slowly injected to create a small hemispherical bleb. If no systemic reactions appear within 15–30 min post intradermal injection of bee venom, the patient is considered negative in the test. In the case of anaphylaxis, the patient should receive intensive medical care, which involved adrenaline and/or dexamethasone injection. Because bee venom sensitivity varies from one person to another, there is no generalized protocol for all patients subjected to bee venom therapy. For sensitive persons, a specialized protocol should be applied. In the case of chronic inflammatory disorders, about 12–20 sessions should be given to get the best results. Some cases require long periods of treatment as in rheumatoid arthritis, psoriasis, multiple sclerosis, and lupus. Patients are advised to obey the following instructions to get the best results [88].

2–3 mg of vitamin C per day is recommended allover bee venom therapy course;650 mg of acetaminophen can be taken in case of fever and chills;Alcohol is strictly prohibited during bee venom therapy;Ice pack can be applied to the site of injection if local systemic reaction occurred (such as swelling and/or scratching);If local reactions occurred (anaphylaxis), the patient should receive an injection of subcutaneous epinephrine (adrenaline) immediately, and then be transferred to hospital.

Bee venom application in human therapies is contraindicated in many conditions such as children under 5 years old, pregnancy, breast feeding, acute and chronic infections, post vaccinations, renal and hepatic failure, cardiac and respiratory problems chronic tuberculosis, hepatitis, acute cancer, and type 1 diabetes mellitus [88].

#### 3.3.2. Route of Administration of Bee Venom

Owing to the protein nature of bee venom, administration by oral route is difficult, as gastrointestinal enzymes can digest it [4]. Bee venom therapy can be administered in different ways such as bees directly stinging specific points, injecting a purified and sterile bee venom, bee venom ointments, creams, pills, drops, Apis homeopathic preparations, electrophoresis, and phonophoresis (Figure 5). Therapy by direct sting using honeybees is considered as a traditional method of treatment. It has multiple disadvantages like pain and inflammation caused by the sting, difficulty for maintaining its regular concentrations in blood, the need for long-term administration of a series of stings or injections because of the short half-life of melittin, and the inconvenience to patients [89]. The relatively short plasma half-life of bee venom and the problematic nature of determining its definite dose have led investigators and practitioners to promote and develop other alternatives, such as the combination with polymers or nanoparticles (NPs) [90].

#### 3.3.3. Bee Venom Therapy in Hospitals

The most known experience of bee venom therapy was in Russian hospitals. Table 1 shows the experience of Dr Krylov and his colleagues [91] in using bee venom therapy in the treatment of multiple sclerosis in Chelyabinsk center of multiple sclerosis in Russia. Moreover, Russian Apitherapist, Dr. Ludyanski, treated different ailments [73] in a Russian hospital using bee venom (Table 2).

#### 3.3.4. Adverse Effects of Bee Venom

Although the therapeutic effects of bee venom have been shown, its safety is still a strong limiting consideration [2]. The adverse effects of bee venom range from mild skin reactions that recover after a few days to life-threating severe or fatal anaphylactic responses. In 2015, a systematic review and meta-analysis evaluated the adverse effects of bee venom therapy. Among the 145 studies, adverse events occurred in 58 studies [64]. These adverse events include immunological systemic reactions, local itching or swelling, allergy, pain, skin problems, and nonspecific reactions [64]. In addition, acute anaphylactic shock, hemolysis, Guillaume–Barre syndrome, and irreversible ulnar nerve injury may occur [86]. The adverse effects of bee venom might be attributed to the hypersensitivity [92,93], persistent local inflammation [94], frequency of bee administration, and venom concentration [2].

In contrast, Kim et al. [95] found that subcutaneous injection of bee venom and its fractions did not have any significant side effects on the general physiological functions of the central nerves, as well as cardiovascular respiratory and gastrointestinal functions at the highest dose tested (200 times and 100 times higher doses than that used clinically, respectively). According the results of such a study, the doses of bee venom, in the therapeutic range or higher, are safe in clinical studies and can be used as safe anti-nociceptive and anti-inflammatory agents.

It is well known that both honey-bee sting and bee venom injection cause pain in addition to inflammation [96]. Much accumulating evidence supports the hypothesis that melittin is the major pain-producing component of bee venom [97]. Chen et al. [97] revealed that subcutaneous injection of melittin causes tonic pain sensation in both animals and humans. This ability to induce pain may be mediated by direct or indirect activation of primary nociceptor cells by melittin.

Other adverse effects of bee venom and melittin include haemolysis because of their high lytic activity on human erythrocyte cells [45,98]. In addition, melittin is cytotoxic for human peripheral blood lymphocytes in a dose- and time-dependent manner. It leads to granulation, morphologic changes, and finally lysis of cells [45,99].

### 3.4. Bee Venom Products

In Western countries, injectable forms of bee venom are an alternative way to some drugs that possess side effects. This is especially true for rheumatoid arthritis. Depending on the type of disease, bee venom can be used in a form of cream [73,100] tablets, or ointments [101]. Other pharmaceutical forms consist of a mixture of bee venom with sterile, injectable fluids and filling them in glass vials or syringes. Moreover, the dry venom is kept lyophilized and then mixed with the solvent at the time of injection [102]. In Europe and China, bee venom solutions are also used with electrophoresis or ultrasonophoresis. Some manufactures added some other bee products (such as pollen propolis, honey, and royal jelly) to bee venom to obtain significant effects. Table 3 shows some products available in markets as well as their dosage form, indications, and manufacturers.

### 3.5. Licensing of Bee Venom as a Drug

There is only one legally approved injectable form of bee in Western European and North American countries, for desensitizing persons who are hyperallergic to bee venom [102]. The bee venom injection was approved as a biotherapeutic medication by South Korean food and drug administration (South Korean FDA), but in the United States (USA), it is waiting the approval of the FDA. There are a lot of known apitherapists in the United States and all over the world that apply apitherapy, such as Bodog F. Beck, M.D. (New York); Raymond Carey, M.D. (California); Stephan Stangaciu, M.D. (Romania); H. O’ Connell, M.D. (Connecticut); Joseph Broadman, M.D. (New York); L.A. Doyle, M.D. (Iowa); Christopher M-H Kim, M.D. (South Korea); Joseph Saine, M.D. (Montreal); and Charles Mraz, Master Beekeeper (Vermont).

### 3.6. Potential of Bee Venom Loaded on Polymers and Nanoparticles (NPs)

Because honeybee stings and bee venom injections have many disadvantages, the design of a suitable sustained release system that provides a long-term and constant therapeutic effect would be necessary. Recently, NPs were prepared from biodegradable polymers such as poly-D, L-lactic-co-glycolic acid (PLGA), alginate, and chitosan. These polymers and their NPs are excellent and effectual carriers for bioactive compounds. Thereby, the bee venom can be loaded on these polymers and their NPs to improve its delivery and release, resulting in improved patient compliance by eliminating the need for frequent injections. The degradation time for NPs can be controlled from days to years by altering the type and amount of polymers, the polymer molecular weights, or the NPs’ structures. Bee venom can maintain sustained release and efficacy for a long time using such polymer NPs, which reduces the administration interval and patient compliance [89,103].

In the same regard, Qiao et al. [104] investigated the interactions between bee venom and the copolymer poly(dl-lactide-co-glycolide-b-ethyleneglycol-b-dl-lactide-co-glycolide) (PLGA–PEG–PLGA). They noticed that the release of bee venom was reduced and the hydrogel was poorly degraded, whereas the exerted biological activities were sustained [104].

In another method of drug delivery, Xing et al. [90] designed a novel formulation for oral administration using coated calcium alginate gel beads-entrapped liposome and bee venom peptides as a model drug for colon-specific drug delivery. In such conditions, the bee venom was protected from being released completely in the stomach and small intestine. The colonic arrival time of the tablets was found to be 4–5 h. The results clearly demonstrated that the coated calcium alginate gel beads-entrapped liposome is a potential system for colon-specific drug delivery.

In 2018, Lee and his colleagues [105] evaluated the efficacy of chitosan/alginate NPs for encapsulating bee venom and their potencies against porcine reproductive and respiratory syndrome virus. In their study, Lee and his colleagues found that nasal-derived NPs were capable of inducing T helper cell type 1 (Th1) immune response and increasing the production of cluster differentiation (CD^4+^), T lymphocytes, memory T cell populations, cytokines (interferon- γ (IFN-γ) and IL-12), and the transcriptional factors such as signal transducer and activator of transcription 4 (STAT4) and **T**-box expressed in **T** cells (T-bet). Likewise, these nasal-derived NPs produced a decrease in the T regulatory cells, cytokines (IL-10 and transforming growth factor-β (TGF-β)) and the transcriptional factors (signal transducer and activator of transcription 5 (STAT5) and Forkhead box protein P3 (Foxp3)).

Alalawy et al. [103] found that loading bee venom on nano-fungal chitosan augmented anti-cancer bioactivity against cervix carcinoma (HeLa) cells and could induce serious apoptosis signs in HeLa cells in a time- and dose-dependent manner.

Researchers at the Washington University in St. Louis have shown that NPs carrying bee venom effectively destroy HIV while leaving surrounding cells unharmed [106]. Moreover, nanofibers loaded with bee venom and propolis exhibited broad spectrum antibacterial activities [45].

## 4. Conclusions and Future Perspectives

Bee venom and its components have a wide range of biological and pharmacological activities. It has been subjected for clinical applications by apitherapists to treat many diseases. It has been licensed for human therapy in different countries and is now available in markets in different forms. The loading of bee venom and its component melittin with polymers and their nano-forms maintains long-term release and enhances bee venom efficiency. However, the use of bee venom loaded on NPs is still at the level of experimental and pre-clinical studies. Thus, this study calls for collaborations between researchers and clinicians to assess the safety and efficacy of bee venom and their component peptides loaded on NPs in the treatment of human diseases.

## 5. Materials and Methods

We searched PubMed, Scopus, Google Scholar, and Web of Science for articles published up to June 2021 making use of the following headings and keywords alone or in varied combinations: “Bee Venom”, “In vitro studies”, “In vivo studies”, “Pharmacological Effects”, “Clinical Application”, “Acupoints”, “Acupuncture”, and “Apitherapy”. The articles chosen for inclusion to gather and collect information in the review article include peer-reviewed articles that have a digital object identifier (DOI) and international standard serial number (ISSN) as well as book chapters with an international standard book number (ISBN). The exclusion criteria include non-peer reviewed literature, articles where the full text was not available, articles unrelated to bee venom, articles with limited information, articles duplicated between databases, conference abstracts, and retracted articles.

## Figures and Tables

**Figure 1 molecules-26-04941-f001:**
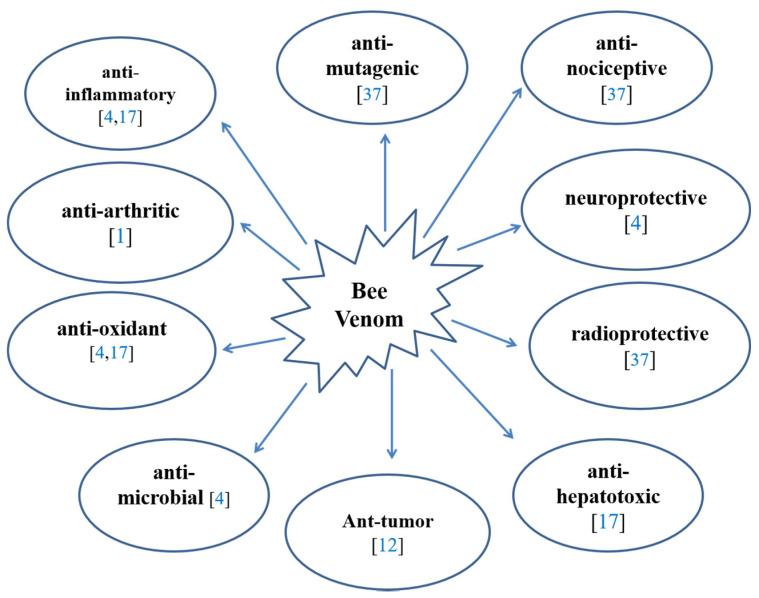
Pharmacological activities of bee venom.

**Figure 2 molecules-26-04941-f002:**
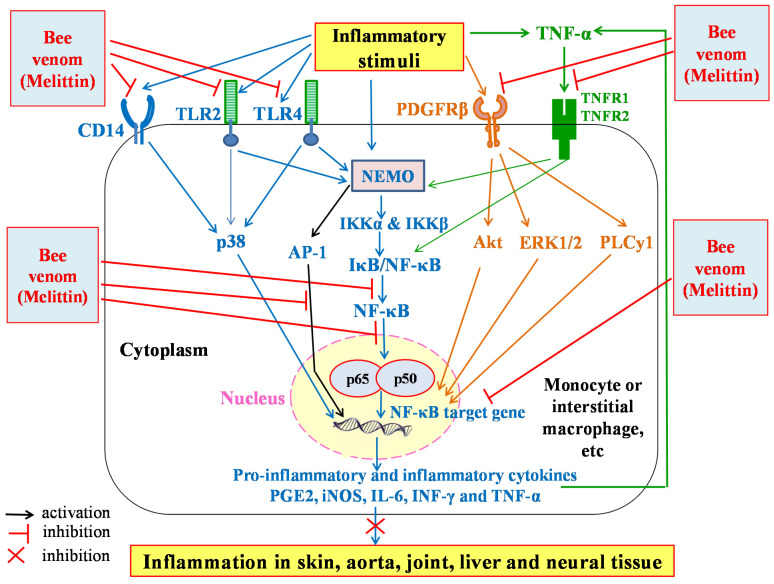
Schematic figure showing the proposed anti-inflammatory mechanisms of action of bee venom and melittin. NF-κB, nuclear factor-κB; IκB, inhibitor of NF-κB; IκB/NF-κB, IκB-NF-κB complex; IKKα, IκB kinase α; IKKβ, IκB kinase β; NEMO, NF-κB essential modulator; TLR, Toll-like receptor; CD, cluster of differentiation; PDGFRβ, platelet-derived growth factor receptor beta; AP-1, activator protein 1.

**Figure 3 molecules-26-04941-f003:**
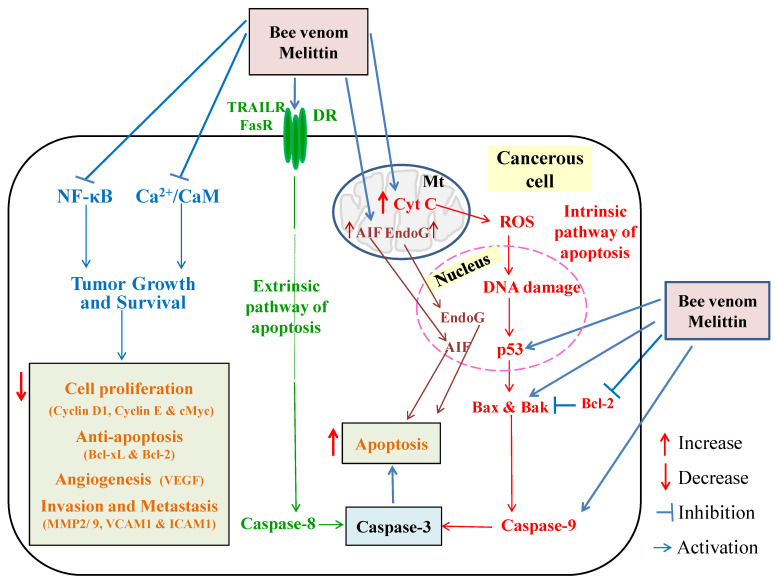
Schematic figure showing the proposed anti-cancer mechanisms of action of bee venom and melittin. DR, death receptor; Cyt C, cytochrome C; ROS, reactive oxygen species; DNA, deoxynucleic acid; p53, protein 53; Bcl-2, B-cell lymphoma 2; Bax, Bcl-2-associated Xprotein; Bak, Bcl-2 homologues antagonist/killer; NF-κB, nuclear factor-κB; Ca^2+^/CaM, calcium-calmodulin complex; VEGF, vascular endothelial growth factor; VCAM1 vascular cell adhesion molecule 1; ICAM1, inter-cellular adhesion molecule 1; MMP2/9, matrix metaloproteinase 2 and 9; cMyc, *c-* Myelocytomatosis; Bcl-xL, B-cell lymphoma extra-large; AIF, apoptosis-induced factor; EndoG, endonuclease G; TRAILR, TNF-related apoptosis-inducing ligand receptor; FasR, fasreceptor.

**Figure 4 molecules-26-04941-f004:**
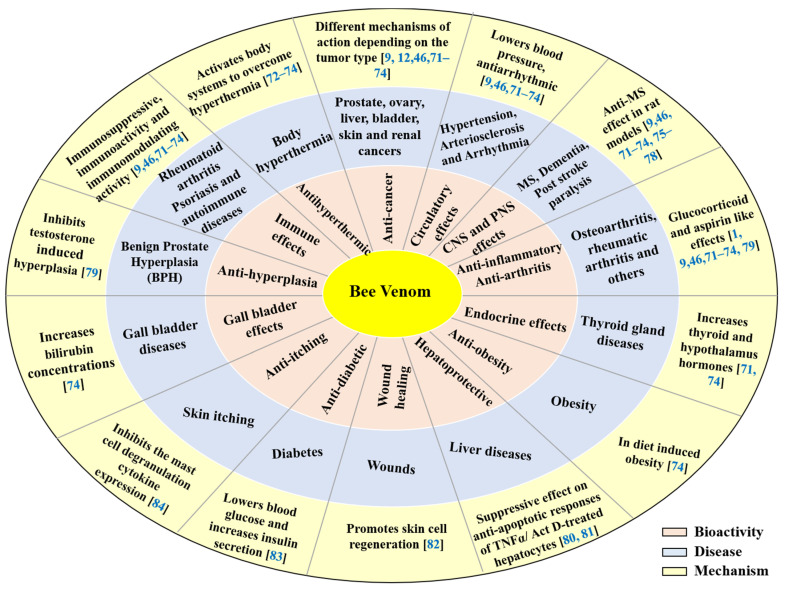
Therapeutic effects and mode of actions of bee venom. Abbreviations: CNS, central nervous system; PNS, peripheral nervous system; MS, multiple sclerosis; BPH, benign prostate hyperplasia; TNF-α, tumor necrosis factor-α; Act D-treated hepatocytes, actinomycin D-treated hepatocytes.

**Figure 5 molecules-26-04941-f005:**
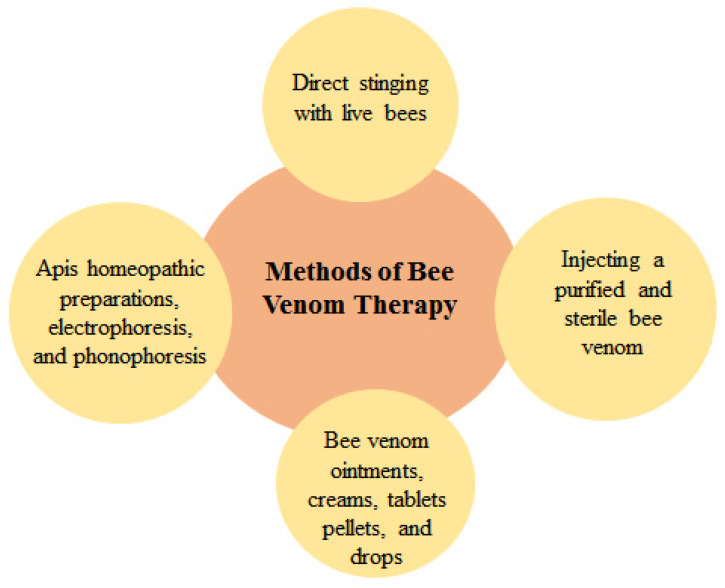
Methods of bee venom therapies.

**Table 1 molecules-26-04941-t001:** Bee venom therapy experiment on patients with multiple sclerosis in Chelyabinsk Russian center [91].

Degree of Response	Number of Patients	Total Improvement (%)	Stop of Demyelination (%)	Remyelination (%)
Primary	24	66	36	29
Secondary	53	84	80	72
Relapse	36	91	85	83

**Table 2 molecules-26-04941-t002:** Dr. Ludyanski’s experience with bee venom in treating different ailments in a Russian hospital [72].

Ailment	No. of Patients with very Good Improvement	No. of Patients with Good Improvement	No. of Patients with No Improvement
Bronchial asthma	38	17	10
Multiple sclerosis	103	72	35
Polyneuritis	22	9	6
Polyarthritis	77	18	15
Trigeminal neuralgia	16	7	2
Poststroke paralysis	196	10	31
Osteochondrosis	1542	352	116
Hypertension	18	9	18
Myopathy	54	8	16
Ganglion nerve inflammation	11	4	1
Inflammation of facial nerve	128	6	1
Cerebellar ataxy	12	7	2
Syringomyelia	140	31	11
Trigeminal neuralgia	16	7	2
Post-traumatic inflammation of the plexus nerve	206	46	21
Arachinoid inflammation	275	20	20

**Table 3 molecules-26-04941-t003:** Some products of bee venom available in markets.

Product	Dosage Form	Indication	Manufacturer
Bee venom mist essence	Spray	Skin protectant	Seoul cosmetics Co. Ltd.
Bee venom eye rescue	Drops	Allergies, red itchy eyes, dry eye, corneal scratches, conjunctivitis, cataracts, tired “computer” eyes.	Bee pharm
Bee venom	Capsules	Support joint health and mobility	Deepbluehealth
Bee venom moisturizer	Cream	For naturally younger looking skin	rodial
Bee venom super serum	Serum	Anti-aging serum	rodial
ApiVENZ	Chewable tablets	Support joint health and mobility	ApiHealth NZ Ltd.

## Data Availability

The data presented in this study are available on request from the corresponding author.

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
