# Peer review of "Bee Venom: From Venom to Drug"

_molecules, 2021, doi:10.3390/molecules26164941_

Round 1
Reviewer 1 Report
The Manuscript entitled "Bee venom: From venom to drug" is very interesting. Briefly, it describes that bee venom and bee-derived toxins can be used in therapy. The subject is very important, no doubt about it, so I appreciate the study. However, important issues are not explored. See my comments in the end.
The English is fine, just few sentences are downright beyond comprehension. See some corrections below:
Minor issues:
Line 30 – review the first sentence. Multiple bee stings can be very scary. Write in a different way.
Line 70 – hymenopteran venoms are much more complex than what was presented. Please see https://doi.org/10.3389/fimmu.2019.02090
Line 78 – Delete complicates, seems weird.
Line 85 – bee venom and bee-derived toxins (add that)
Line 92 – Add more references that support that.
Line 94: bitter taste? Someone tried it?
Lines 98-112 – add the citations after each observed activity.
Line 109 – and is not in italics
Lines 111 – 113 – replace comma and “and”.
Line 119 – injected rabbits with – it would be better injected into rabbits…..
Line 121 – detail the abbreviations GST and GSH. – see line 129 as well
Line 133 – by a decrease …. and an increase…
Line 155 – Please see discuss the synergism effect of melittin and PLA2 to induce cell lysis. See: https://doi.org/10.3389/fphar.2020.00611
Line 167 – Where are these promising results? I believe that this section should be improved. There many others studies in vitro with bee venom and derived toxins.
Line 174 – Add a ref after stinger.
Lines 178 – 182 – I believe that the methods deserve a figure illustrating that since this is a review. Author should also discuss that bees will die after direct sting.
Lines 184 – 187 – please explore how it improve the patient’s disease/symptoms such as authors did in paragraph from line 192.
Line 220 – so long section title for a small section.
Line 230 – is is lacking the verb (are injected)
Line 266 – Dr????
Figures:
Figure 1 is very poor. Please make it more illustrated or delete it.
Figure 2: the content of it enriches a lot the review. But I still think that authors can improve it in style. Please make some efforts to do that.
Tables 1 and 2 – add the ref.
Major problem:
Although I recognize that authors performed a nice work searching for different approaches and studies that used bee venom or its toxins in therapy (especially in Fig. 2), they did not explore the mechanism of action (even hypothetical) and (in most of the review) how the therapy benefited the patients or relieve their symptoms.
Besides that, it is missing an entire section about the apitherapy side effects (authors mentioned some in a superficial way). For instance, the melittin-induced pain should be explored.
Author Response
Reviewer 1 and reviewer 2 comments and author response were found in attached table.

Reviewer 2 Report
Most of animal venoms are complex cocktails of bioactive compounds. Basically, they are mixture of proteins and peptides, salts and organic components (e.g. amino-acids and neurotransmitters). In the last years there has been a growing interest in animal venoms because they display beneficial effects towards several disorders and pathological conditions. Though, the most widespread and frequently lethal venomous animals encountered by humans are the snakes many invertebrate animals (e.g. sea urchins, cone snails, insects up to jellyfish and sponges) are capable to synthesize poisonous venoms but with beneficial pharmacological effects. In the review titled "Bee venom: from venom to drug" the authors attempted to summarize the potential of bee venom, either as a such or its fractions, in treating human disorders and counteracting drug toxicity. On the whole I appreciated the manuscript however before to be considered for publications there are few minor issues that need to be sorted-out to improve the quality of the review. 1) It would be highly appreciated to implement a little bit the sections about the biochemistry of bee venom, the molecular mechanisms of action and I recommend also to add a figure summarizing the effects of been venom at cellular, subcellular and molecular levels; 2) It would be advisable shrink a little bit the very first part of the section "Introduction"; 3) References should be slightly trimmed and restricted to the English language peer-reviewed articles (please skip youtube video); 3) There are few typos that need to be edited e.g. page 7 line 266 "...experience of Dr. [80] of bee venom therapy...." 4) Acronyms should be stated in-extenso, at least the first time, e.g. page 3 line 121 what does MDA stand for?Author Response
Reviewer 1 and reviewer 2 comments and author response were found in the attached table
